# Response to Anti-PD1/L1 Antibodies in Advanced Urothelial Cancer in the ‘Real-Life’ Setting

**DOI:** 10.3390/ph15091154

**Published:** 2022-09-16

**Authors:** Moran Gadot, Ido Arad, Eshetu G. Atenafu, Meital Levartovsky, Orith Portnoy, Tima Davidson, Rachel Schor-Bardach, Raanan Berger, Raya Leibowitz

**Affiliations:** 1Sheba Medical Center, Oncology Institute, Tel-Hashomer 52621, Israel; 2Department of Biostatistics, Princess Margaret Cancer Centre, Toronto, ON M5G 2M9, Canada; 3Sheba Medical Center, Diagnostic Imaging Department, Tel-Hashomer 52621, Israel; 4Sackler Faculty of Medicine, Tel-Aviv University, Tel-Aviv 699781, Israel; 5Department of Nuclear Medicine, Sheba Medical Center, Tel-Hashomer 52621, Israel; 6Shamir Medical Center, Oncology Institute, Zerifin 70300, Israel

**Keywords:** immune checkpoint inhibitors, metastatic urothelial carcinoma, PD-L1, PD-1, response, lymph nodes

## Abstract

Immune checkpoint inhibitors (ICIs) are now the standard of care for metastatic urothelial carcinoma (mUC) patients. Our aim was to describe the activity of ICIs in mUC and find the clinical parameters associated with response. This is a retrospective, single-center chart review of mUC patients receiving ICIs. The overall survival (OS) was plotted using the Kaplan–Meier method and was compared using a log-rank test. Associations between the variables and responses were analyzed by univariate and multivariable analyses, using either logistic regression or a Chi-square/Fisher’s exact test. Ninety-four patients received ICIs, 85% of which were in the second line or beyond; the median age was 71.8 years, and 82% were men. Six (6.4%), 11 (11.7%), 7 (7.4%) and 70 (74.5%) patients achieved a complete response (CR), partial response (PR), mixed response/stable disease (M/SD) or progressive disease (PD), respectively. The median overall survival was 3.2 months for the entire cohort and was significantly different according to the response pattern—not reached, 32.3, 6.4 and 2.0 months for CR, PR, M/SD and PD, respectively. The response was not significantly associated with the line of treatment. ‘Site of metastasis’ was associated with the response, and the absolute neutrophil count was borderline associated with the response. In summary, we found a substantial variance in the potential benefit from ICIs in mUC, emphasizing the need for predictive biomarkers and frequent monitoring of mUC patients receiving ICIs.

## 1. Introduction

After decades in which there were no advancements in the treatment of metastatic urothelial cancer (mUC), the last years have seen the introduction of immune checkpoint inhibitors (ICIs) into the arsenal of active drugs for this disease. Following the initial report on the activity of pembrolizumab in five different types of cancer, presented in abstract form at the end of 2014 (and later published) [1,2,3,4,5], medium phase-II and large phase-III clinical trials were quickly initiated for urothelial cancer, assessing anti-PD1 and anti-PDL1 in the advanced, platinum-refractory setting [6,7,8]. Ultimately, the anti-PD1 antibody pembrolizumab [8], but not the anti-PDL1 antibody atezolizumab [7], was shown to be superior to second-line chemotherapy, despite having response rates of around 20%.

Pembrolizumab and atezolizumab received accelerated approval for frontline use in cisplatin-ineligible patients based on the surrogate endpoints from two phase-II single-arm trials [9,10]; however, their use was later restricted by regulatory agencies and given only to cisplatin-ineligible UC patients who were PD-L1 positive or patients who were ineligible for any platinum-containing chemotherapy [11].

In line with these prospective observations, a retrospective cohort study of more than 200 mUC patients receiving first-line or carboplatin-based chemotherapy found a lower OS rate at 12 months for ICI but a higher OS rate at 36 months relative to carboplatin-based chemotherapy [12].

It has been repeatedly established that patients in the ‘real-life’ setting generally fare worse than patients in clinical trials across all common oncological outcomes (such as response, progression-free survival (PFS) and overall survival (OS)) for several reasons [13]. Our aim in this current analysis is to describe the characteristics of response and survival in urothelial cancer patients receiving anti- PD1/L1 antibodies at the Sheba medical center, and to characterize the clinical or biochemical variables associated with responses to treatment.

## 2. Results

Between November 2014 and March 2019, 94 patients received ICIs for urothelial cancer at the Sheba medical center. The median age was 71.8 (range 43.7–95.7) years, and 77 (82%) were men. The primary disease site was the bladder in 77% of the cases, the upper urothelial tract (ureters and renal pelvis) in 17%, and in 6%, it presented in both sites concurrently. Approximately one-third of the patients (35%) presented with metastatic disease, whereas the remaining patients initially presented with localized disease. In 85% of the patients, the cancer was pure urothelial, in 9%, it was a mixture of pure urothelial with other histological types, and in 4% of patients, the cancer was predominantly squamous. The metastatic site included the lymph nodes (LNs) as a single site in 11%, visceral organs (liver, lung, bladder, pelvis, kidney and peritoneum, with or without LNs) in 61%, and bones (with or without additional site involvement) in 29%.

A total of 61% of patients received ICIs in the second line following a platinum-based combination therapy; 24% of the patients received ICIs in the third line or above, and 14% of patients received ICIs in the first line (mainly due to an inability to receive chemotherapy). Almost half of the patients (47%) in our cohort had a significantly compromised ECOG PS of 3, whereas about a quarter (28%) had a compromised ECOG PS of 2, and another quarter (26%) had a preserved ECOG PS of 0/1. The patient characteristics and baseline lab parameters are given in Table 1.

The treatment characteristics are given in Table 2. A little over two-thirds of the patients (71%) received pembrolizumab, whereas the remaining patients received any of the three other ICIs (durvalumab, nivolumab, atezolizumab). Four patients received a combination of anti-PD1/L1 antibody and an additional investigational ICI as part of a clinical trial. The median number of treatment cycles was 3 (range 1–43). The reasons for discontinuation included death or progressive disease in 80%, toxicity in 7% and the achievement of a complete response in 4%. In four patients for which the treatment was discontinued due to the achievement of a complete response, the median number of administered cycles was 27 (range 10–43). Further, 9% of patients were still on treatment at the time of the data analysis.

A total of 79 patients (84%) had either no side effects or only fatigue; 10 patients (11%) had two irAEs, and 5 patients (5%) had three or more. Approximately a third of the patients (33%) received steroids to ameliorate irAEs throughout the treatment.

The frequency of irAEs is given in Figure 1. A little more than 10% of patients had a rash on treatment, a little less than 10% had colitis, and approximately 5% of patients developed pneumonitis. Toxicity in the CNS occurred in 2.8% of patients (mainly encephalitis). Additional rare irAEs included myositis in one patient, peripheral neuropathy in one patient, and hyponatremia (that was not a manifestation of hypocortisolism) in one patient.

The response characteristics are given in Table 3. Six (6.4%) patients achieved CR, 11 (11.7%) patients achieved PR, 7 (7.4%) had a mixed response or stable disease, and 70 (74.5%) had progressive disease as their best response. The overall response rate (calculated as the sum of CR+PR) was 18.1%. The overall survival for the entire cohort was 3.2 months (range 0.3–over 44.5 months), and the median duration of the response was 2.1 months (range 0.3–over 34.9 months).

The median OS was not reached for patients achieving CR (as all six patients were still responders at the time of analysis and, hence, were above 48 months); 32.3 months for patients with PR; 6.4 months for patients with mixed/SD; 2 months for patients with PD. The survival curves of the four response groups are seen in Figure 2, with the difference between the groups highly statistically significant (*p* < 0.0001 using the log-rank test). There was also a strong association between ECOG PS and the median overall survival: 7.0 months for patients with ECOG PS = 0/1; 1.9 months for patients with ECOG PS = 2; 1.3 months for patients with ECOG PS = 3/4 (*p* < 0.05 using log-rank test). Nonetheless, there were three patients in each group of the compromised ECOG PS (corresponding to 13% and 15% of the ECOG PS = 2 and ECOG PS = 3/4 groups, respectively) that survived more than a year following an ICI start.

We next assessed the association between the clinical and biochemical parameters and response using univariate analysis. The following parameters were not significantly associated with the response: age, primary site, histology, metastatic disease at presentation, the previous number of treatment lines, baseline platelet or lymphocyte count, hemoglobin, lactate dehydrogenase (LDH) levels, ECOG performance status (PS), neutrophil-to-lymphocyte ratio (NLR) and sex. Two parameters were associated with response, with a p-value of 0.15 or less—the metastatic site (*p* = 0.005) and the absolute neutrophil count (*p* = 0.15). We changed the continuous variable ‘neutrophil count‘ to a dichotomous scale of neutrophils above or below the median, which was calculated to be 4.6 K/microL. This binary variable was borderline associated with the response in the univariate analysis (*p* = 0.06; Table 4).

We then performed a multivariable analysis using two variables: ‘metastatic site’ and the binary variable ‘neutrophil count above or below median’. The site of metastasis retained a statistically significant association with the response: patients with metastasis solely in the LNs had a response rate of 60%; patients with visceral metastasis had a response rate of 14.8%; patients with bone metastasis had a response rate of 8.7% (*p* < 0.05; Table 5 and Table 6). The ‘neutrophil count above median’ was borderline associated with the response (*p* = 0.056; Table 5).

## 3. Methods

### 3.1. Patients

This is a retrospective chart review of all patients who received any type of anti-PD1 or anti-PD-L1 antibody (designated collectively as ‘immune checkpoint inhibitors’ (ICIs)) for non-resectable/metastatic urothelial cancer of any primary site (bladder or upper tract). Patients were included if they had any histology of bladder cancer and received ICIs at any line of treatment, as long as it was not given concurrently with chemotherapy or as maintenance following the non-progression of a previous line. Four patients (4%) who received a combination of ICIs, one of which was an anti- PD1/L1 antibody (as part of a clinical trial), were included. Patients who were treated with ICIs as a ‘re-challenge’, i.e., following a previous course of such treatments, were only assessed for drug activity upon their first exposure; however, they were included in the overall survival analysis. Following the institutional review board’s ethical approval (#SMC-4963-18), clinical data and lab results were extracted from electronic medical records. Baseline lab tests were defined as those taken within two weeks prior to the first course of ICI.

### 3.2. Endpoints

There were two endpoints to this study. The first was to quantify the response patterns and calculate the overall survival according to the type of response achieved. The second was to find the clinical and biochemical parameters associated with the response.

### 3.3. Definitions of Response and Toxicity

Complete response (CR) was defined as the complete disappearance of metastasis on computed tomography (CT) or magnetic resonance imaging (MRI) scans at any timepoint from the initiation of ICI treatment. Partial response (PR) was defined as a clinically meaningful decrease in the extent of metastasis based on physicians’ discretion. These criteria were chosen rather than the formal RECIST criteria since, in ‘real-life’, responses can clearly be clinically meaningful, even if a 30% decrease in the sum of diameters is not observed. Progressive disease (PD) was defined as any of the following: unequivocal radiological progression (with or without subsequent confirmatory scans), unequivocal clinical deterioration, or death. A mixed response was defined when some of the metastatic lesions decreased in size and some increased, and stable disease (SD) was defined when neither a clinically meaningful decrease nor an unequivocal increase in the size of metastatic lesions occurred. ‘Response’ was defined as achieving either CR or PR, and ‘non-response’ was defined as either SD, mixed response or PD.

Adverse events were defined as immune-related (designated ‘immune related adverse events (irAEs)) by the treating medical oncologist following standard diagnostic procedures and to his/her best clinical judgment. The irAEs either belonged to one of the frequent categories—rash, colitis, nephritis, hepatitis, pneumonitis, endocrinopathies and immune-mediated central nervous system (CNS) toxicity—or were defined as ‘other’.

### 3.4. Statistical Analysis

We plotted the overall survival for the 4 different response groups using the Kaplan–Meier method and calculated the statistical significance using the log-rank test. The potential variables associated with the response were analyzed by univariate analysis, using either logistic regression for the continuous variables or χ2/Fisher’s exact test (depending on the expected cell frequency) for the categorical variables. Multivariable analysis was carried out using logistic regression on all variables, with a *p*-value of <0.07 on the univariate analysis. All tests were two-tailed, with a probability of <0.05 considered statistically significant. Statistical analyses were performed using version 9.4 of the SAS System for Windows (Copyright © 2022-2012 SAS Institute, Inc., Cary, NC, USA) and the open-source statistical software R version 3.6.3 (R Core Team, R Foundation for Statistical Computing, Vienna, Austria, 2020).

## 4. Discussion

In this retrospective analysis of a very heterogeneous group of metastatic UC patients, we found a response rate of 18.1% to ICIs. This rate is similar to the response rates reported in the prospective phase-III trials [7,8], despite the less stringent definition of response in our analysis. It has been repeatedly shown that ‘real-life’ patients fare worse than clinical trial patients [13,14]. Here, it is plausible that the compromised performance status of the patients negatively affected response and survival, as described in [15]. Indeed, we found an association between PS and OS, although there were a few patients with compromised or extremely compromised PS who had prolonged survival following the start of treatment, stressing the complex association between these two parameters concerning ICIs. The patient characteristics in this cohort are very typical of mUC patients, and we conjecture that this cohort’s performance status is indeed reminiscent of that seen in ‘real-life’ patients with this disease. Of note, there was a very low rate of disease stability or mixed response to the ICIs, and most patients progressed. The median number of the treatment cycles—three—indicates that most patients stopped their treatment prior to their fourth cycle and less than 12 weeks after treatment initiation.

There was a very strong association between the type of response and OS, with patients achieving a CR having an excellent prognosis and those achieving PD having an extremely dismal prognosis, with a median OS of merely 2 months. A commentary analyzing the updated results of the KEYNOTE-045 trial of pembrolizumab in the second line in UC patients [16] pointed out that different parts of the mUC population get markedly different levels of benefit (or lack thereof) from pembrolizumab [17]. Our retrospective results strongly support this claim and suggest that frequent clinical monitoring should be practiced within the first few months of ICI treatment and that a high index of suspicion for rapid progression should be exercised. The vast variance in the extent of benefit from using ICIs is of great concern, as the rapid clinical deterioration seen in non-responders may preclude offering these patients additional lines of treatment following ICI failure, as has indeed been reported by Lista et al. [18]. Vigilance and careful monitoring must be exercised, especially in lieu of the clear trend to expedite the administration of ICIs to the first-line setting in cisplatin-ineligible patients [19]. Moreover, it has been suggested that some patients with solid malignancies have ‘hyper-progression’ on ICI [20,21,22,23,24], namely an increase in the rate of disease progression on treatment. Clearly, our retrospective analysis cannot confirm or negate the existence of such a response type, necessitating prospective clinical trials to assess this very point. Based on the literature and our clinical experience, our current practice is to inform patients of the possibility of rapid disease progression following the initiation of ICIs and to perform frequent clinical and laboratory assessments prior to a formal radiological disease assessment.

We found an association between the site of metastasis and response to ICI, with a significantly higher response in patients whose metastases are confined to lymph nodes and a significantly lower response in patients whose metastases are confined to the bone. Urothelial cancer without liver or bone metastases has been known for decades to portend a better prognosis and response to chemotherapy [25]. Conversely, the existence of bone metastasis was found to be associated with a worse response to cisplatin-containing combination chemotherapies [26]. Our analysis may also suggest that a higher neutrophil count is associated with a worse response to ICIs, although this association is only borderline significant. Neutrophilia is a validated prognostic factor in metastatic renal cell carcinoma [27], uterine carcinosarcoma [28], stage III NSCLC [29] and locally advanced rectal cancer [30] but has not been suggested as a prognostic or predictive factor of response to ICIs to date, to the best of our knowledge. A low neutrophil-to-lymphocyte (NLR) count has repeatedly been shown to be associated with a response to chemotherapy in several settings of urothelial cancer [31,32]. Clearly, our observation needs further validation in larger cohorts.

The main limitation of our work is the heterogeneity of the cohort, mainly in terms of PS and in terms of lines of treatment, and our observations must be taken with this significant limitation in mind. Nonetheless, the consistency of our observation with previously published prospective and retrospective trials is reassuring and lends strength to our proposition that mUC patients receiving ICIs should be monitored frequently.

## 5. Conclusions

Our recent analysis highlights the significant variance in the response and consequent potential survival benefit (or lack thereof) from using ICIs for urothelial cancer. Our work points to sole LN involvement as a potential predictor of response, to be further assessed prospectively. The association between a high neutrophil count and lack of response should also be further studied. Hopefully, novel clinical factors, together with the emerging molecular factors of both the tumor and the micro-environment, will help better define the mUC patients most likely to respond to immune checkpoint inhibitors.

## Figures and Tables

**Figure 1 pharmaceuticals-15-01154-f001:**
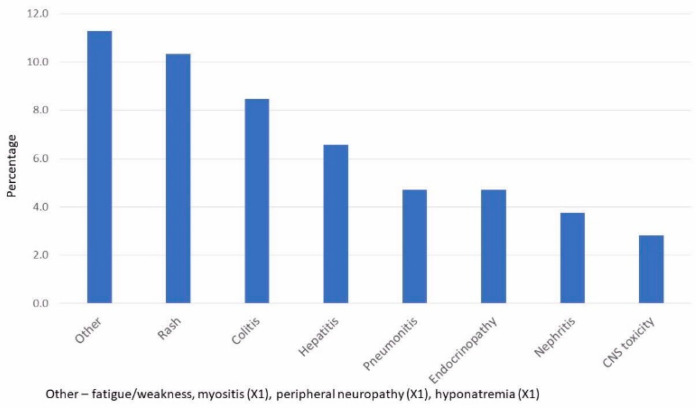
Frequency of Immune-Related Toxicities.

**Figure 2 pharmaceuticals-15-01154-f002:**
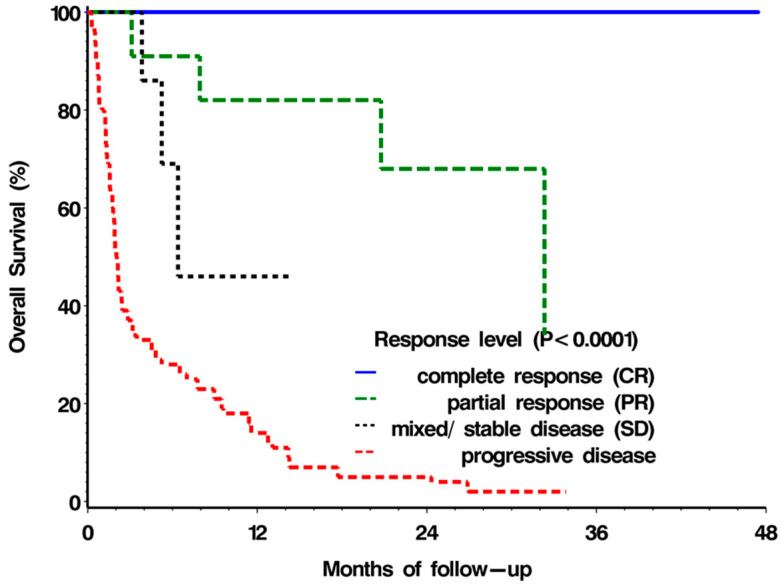
Overall survival according to response type.

**Table 1 pharmaceuticals-15-01154-t001:** Patient Characteristics.

Patient Characteristics		N	%
Sex	Male	77	82
	Female	17	18
Histology	Pure urothelial	80	85
	Mixed	8	9
	Predominantly squamous	4	4
	Other	2	2
Primary site	Bladder	72	77
	Upper tract	16	17
	Bladder + Upper tract	6	6
Sites of metastasis	Visceral (soft tissue in pelvis or metastatic)	57	61
	LNs only	10	11
	Bones (with or without other sites)	27	29
Metastatic at presentation	No	61	65
	Yes	33	35
Definitive treatment	Surgery	34	36
	RT	13	14
	Surgery + RT	2	2
	Not applicable/unknown	44	47
Prior neo-adjuvant/adjuvant chemo	Yes	28	30
	No	66	70
# of previous chemotherapy lines	0	14	15
	1	57	61
	2 or more	23	24
ECOG PS	0/1	24	25
	2	26	28
	3/4	44	47
	**Characteristic**	**Median**	**Range**
	Age (years)	71.8	43.7–95.7
	Hemoglobin (gr/dL)	10.3	7–15.5
	WBC (K/microL)	7.2	2.3–37.3
	Platelets (K/microL)	235	16–668
	Neutrophils (K/microL)	4.6	1.5–35.8
	Lymphocytes (K/microL)	1.1	0.25–6.0
	LDH (IU/L)	218	114–7143
	Alkaline phosphatase (IU/L)	120	37–1993
	Albumin (gr/dL)	3.5	2.1–6.3

**Table 2 pharmaceuticals-15-01154-t002:** Treatment Characteristics.

Treatment Characteristics		N	%
IO type	Pembrolizumab	67	71
	Atezolizumab	19	20
	Nivolumab	3	3
	Durvalumab	1	1
	Anti-PD1 + Other	4	4
Median # of IO cycles	3	Range 1–43	
Reasons for discontinuation	Death	19	20
	Deterioration (clinical and/or radiological)	56	60
	Complete response	4	4
	Toxicity	7	7
	NA-still on treatment	8	9
Received steroids for toxicity	No	62	66
	Yes	32	34

**Table 3 pharmaceuticals-15-01154-t003:** Response Characteristics.

Response Type	N	%	Median OS (95% CI; Months)
Complete response (CR)	6	6.4	NR
Partial response (PR)	11	11.7	32.3 (>7.9)
Stable disease (SD)/ mixed response	7	7.4	6.4 (>3.8)
Progressive Disease (PD)	70	74.5	2.0 (1.7–2.5)

**Table 4 pharmaceuticals-15-01154-t004:** Univariate Analysis of Clinical Variables Associated with Response.

	Level	*p*-Value	OR	95% CI	
				Lower	Upper
Worst metastatic Site	node only vs. bone	0.005	15.75	2.30	107.93
	visceral disease vs. bone		1.82	0.36	9.13
Neutrophil (K/microL)		0.15	0.87	0.72	1.05
Neutrophil > 4.6 (K/microL)	Yes vs. No	0.06	0.34	0.11	1.06

**Table 5 pharmaceuticals-15-01154-t005:** Multivariable Analysis of Clinical Variables Associated with Response.

Parameter	Level	*p*-Value	OR	95% CI	
				Lower	Upper
Worst metastatic Site	node only vs. bone	0.007	21.6	2.6	180.0
Neutrophil > 4.6 K/microL	Yes vs. No	0.054	0.28	0.08	1.02

**Table 6 pharmaceuticals-15-01154-t006:** Response Rates and Odds Ratio (OR) by Clinical Characteristics.

Clinical Characteristic		Response Rate	95% CI
Metastatic site	Visceral	14.8	(7–26)
	Bone	8.7	(1–28)
	LN only	60.0	(26–88)
Neutrophils (K/microL)	>4.6	12.2	(4–26)
	≤4.6	29.3	(16–46)

## Data Availability

Data is found within the article.

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
