# Peer review of "Response to Anti-PD1/L1 Antibodies in Advanced Urothelial Cancer in the ‘Real-Life’ Setting"

_pharmaceuticals, 2022, doi:10.3390/ph15091154_

Round 1

Reviewer 1 Report

Major comments:

1) The authors should present an overview of the clinicopathological characteristics and the clinical outcome of all mUC patients treated with ICIs. Importantly, they should test whether ICI-responders express PD1 and/or PD-L1 higher than the non-responders. The Figure should also show the duration of each patient's treatment and response (according to RECIST). 

2) It would be worth investigating the association between patient response to ICIs and the immune subtype of the tumor (Thorsson V, et al. 2018. PMID: 29628290): C1 (wound healing); C2 (IFN-gamma dominant); C3 (inflammatory); C4 (lymphocyte depleted); C5 (immunologically quiet); C6 (TGF-b dominant)

3) Why did the authors not monitor any immune-related adverse events in the ICI-treated patients?

Minor comments:

1) In the "patients" paragraph of Methods, please correct "anti-PD1L" with "anti-PD-L1"

2) In the Conclusions section, please begin the sentence "our recent analysis" with a capital word ("Our").

3) Please correct the author names in references #10 & #11 and delete the http link.

Author Response

Major:

1. An overview of the clinic-pathological characteristics of the patients is given in the first 2 paragraphs of the ‘results’. An overview of the response of the entire cohort is given in the third paragraph, to which we have now added (p.5) the overall survival and duration of response of the entire cohort -  “The overall survival for the entire cohort was 3.2 months (range 0.3->44.5 months), and the median duration of response was 2.1 months (range 0.3->34.9 months)”. We also added a brief sentence describing the median OS of the entire cohort to the abstract.

As for PD1/L1 status – this data is not readily available in our medical records as these tests are not routinely performed for patients receiving ICIs in the second line setting in Israel. This is why this was not part of our research aim. Of note, the relationship between PD1/L1 expression and response to ICIs has been extensively studied in both prospective and retrospective analyses and we do not think that our relatively small cohort can shed novel light on the matter. Our contribution is rather in showing that there are additional clinical parameters associated with response, that have not yet been suggested (neutrophil count and metastatic site).

2. The answer here is similar – we do not have data regarding the molecular phenotype of the tumors as this is not routinely done in Israel. The IRB approval we received for this specific analysis did not include a genetic component, and thus retrospective analysis of the samples was not feasible as part of this research. Again – ours is a ‘bread-and-butter’ manuscript describing a ‘real- life’ cohort, which generally lacks, in this context, molecular dat; it merely highlights response characteristics and clinical parameters associated with response.

3. Of course we have monitored, collected and described the toxicity data. This paragraph was omitted in a previous round of corrections, based on the comments of a different reviewer. A paragraph describing the method of toxicity collection has been re-introduced into the ‘methods’ section, and the toxicity has now been given in the ‘results’ and provided in Fig. 1 -

“79 patients (84%) had either no side effects or only fatigue; Ten patients (11%) has two irAEs, and 5 patients (5%) has 3 or more. Approximately a third of the patients (33%) received steroids for amelioration of irAEs throughout the treatment.

The frequency of irAEs is given in Fig.1. A little more than 10% of patients had a rash on treatment, a little less than 10% had colitis, and about 5% of patients developed pneumonitis. Toxicity in the CNS occurred in 2.8% of patients (mainly encephalitis).  Additional rare irAEs included myositis in one patient, peripheral neuropathy in one patient, and hyponatremia (that was not a manifestation of hypocortisolism) in one patient”.

All in all, the toxicity was predictable and in the frequency observed in similar studies.

Minor – all minor corrections were made expect in ref. #11, which is an FDA announcement that is not a formal citation and thus given as a URL.

Reviewer 2 Report

The manuscript describes retrospective study of chart reviews of patients with bladder cancer who received ICI (anti-PD1/PDL1). The authors defined included patients' demographics, clinicopathologic parameters as well as disease outcomes and neutrophil counts. I have the following concerns:

1. patient population is heterogeneous which confounds statistical analysis and interpretation of the results and data.

2. Data presentation, analysis, interpretation and discussion of the results is inadequate.

3. It is not clear why the neutrophil count is used as an analysis variable. 

Author Response

1.Indeed, this is 'real-life' cohort and as such, inherently the patient population is heterogenous. This is clearly stated throughout the manuscript. Nontheleess, the statistical tests performed are the appropriate ones, and we were very careful not to over-interpret our results, being very aware of their inherent limitations. The results do show that the majority of patients treated with ICIs progress, and that stability (i.e.-non-response-non-progression) is a rare event with these types of treatment. We have not read a paper stating this so far and so think this presentation is of some novelty. We also found 2 clinical parameters potentially associated with response, to be further studied in prospective clinical trials. Cleary we do not claim that this paper is 'groundbreaking', yet we think it might be an interesting read to the practicing medical oncologist or the translational-researcher. 
Since the reviewer is (rightfully) concerned, , we now further stressed this limitation in the concluding paragraph of the discussion, and further re-assured our readers that our observations are in complete agreement with others – “ The main limitation of our work is the heterogeneity of the cohort, mainly in terms of PS and in terms of lines of treatment, and our observations must be taken with this significant limitation in mind; nonetheless, the consistency of our observation with previously published prospective and retrospective trials is re-assuring, and lends strength to our proposition that mUC receiving ICIs should be monitored frequently”.

2.In the absence of more concrete criticism (other than the general saying that the research design and results must be improved), it is hard to address this point. Still, we have now significantly improved the manucript based on the othere reviewer's comments (including the addition of toxicity data), and hope this will also address this comment. 

3. We assessed all blood test parameters known to be prognostic in this or other diseases, such as LDH, albumin, calcium, hemoglobin, neutorphil count, lymphocyte count, platelet count, etc. Only the neutorphil count was associated with response. This is clearly stated in p.8  in the discussion - "Neutrophilia is a validated prognostic factor in metastatic renal cell carcinoma27, uterine carcinosarcoma28, stage III NSCLC29 and locally advanced rectal cancer30 but has not been suggested as a prognostic or predictive factor of response to ICIs to date, to the best of our knowledge" 

Round 2

Reviewer 1 Report

Although the authors have adequately responded to 2 out of the 3 major reviewer queries (which is due to the fact that they did not hold any molecular data for their patient cohort), their study can be accepted for publication in its present form. The novelty in this work is that there are additional clinical parameters associated with response, that have not yet been suggested (neutrophil count and metastatic site).